# Changes in Alprazolam Metabolism by CYP3A43 Mutants

**DOI:** 10.3390/biomedicines10123022

**Published:** 2022-11-23

**Authors:** Jie Zhao, Sijie Liu, Clemens Alexander Wolf, Gerhard Wolber, Maria Kristina Parr, Matthias Bureik

**Affiliations:** 1School of Pharmaceutical Science and Technology, Tianjin University, 92 Weijin Road, Nankai District, Tianjin 300072, China; 2Pharmaceutical and Medicinal Chemistry (Pharmaceutical Analysis), Institute of Pharmacy, Freie Universitaet Berlin, Koenigin-Luise-Strasse 2+4, 14195 Berlin, Germany; 3Pharmaceutical and Medicinal Chemistry (Computer-Aided Drug Design), Institute of Pharmacy, Freie Universitaet Berlin, Koenigin-Luise-Strasse 2+4, 14195 Berlin, Germany

**Keywords:** alprazolam, CYP3A enzymes, drug metabolism, *Schizosaccharomyces pombe*, CYP3A43 mutants

## Abstract

Alprazolam is a triazolobenzodiazepine which is most commonly used in the short-term management of anxiety disorders, often in combination with antipsychotics. The four human members of the CYP3A subfamily are mainly responsible for its metabolism, which yields the main metabolites 4-hydroxyalprazolam and α-hydroxyalprazolam. We performed a comparison of alprazolam metabolism by all four CYP3A enzymes upon recombinant expression in the fission yeast *Schizosaccharomyces pombe*. CYP3A4 and CYP3A5 show the highest 4-hydroxyalprazolam production rates, while CYP3A5 alone is the major producer of α-hydroxyalprazolam. For both metabolites, CYP3A7 and CYP3A43 show lower activities. Computational simulations rationalize the difference in preferred oxidation sites observed between the exemplary enzymes CYP3A5 and CYP3A43. Investigations of the alprazolam metabolites formed by three previously described CYP3A43 mutants (L293P, T409R, and P340A) unexpectedly revealed that they produce 4-hydroxy-, but not α-hydroxyalprazolam. Instead, they all also make a different metabolite, which is 5-N-O alprazolam. With respect to 4-hydroxyalprazolam, the mutants showed fourfold (T409R) to sixfold (L293P and P340A) higher production rates compared to the wild-type (CYP3A43.1). In the case of 5-N-O alprazolam, the production rates were similar for the three mutants, while no formation of this metabolite was found in the wild-type incubation.

## 1. Introduction

Alprazolam is a triazolobenzodiazepine which is most commonly used in the short-term management of anxiety disorders (such as panic disorder or generalized anxiety disorder), often in combination with antipsychotics. It is also used for the treatment of chemotherapy-induced nausea.

The cytochrome P450 enzymes (CYPs or P450s) are a superfamily of heme-containing monooxygenases that are present in all kingdoms of life and are able to catalyze a broad range of reaction types, including hydroxylations, C=C bond epoxidation, and heteroatom oxygenations [1]. Among the 57 human CYPs, the four members of the human CYP3A subfamily are responsible for the metabolism of a large percentage of drugs [2]. Alprazolam was shown to be metabolized by human CYP3A4 and CYP3A5 to the main metabolites 4-hydroxyalprazolam and α-hydroxyalprazolam [3]. Subsequently, it was reported that the formation of 4-hydroxyalprazolam by CYP3A4 proceeded twofold faster than that by CYP3A5. In contrast, clearance by CYP3A5 was about 3-fold higher than that by CYP3A4 in the case of alpha-hydroxylation [4]. Later studies showed that CYP3A7 and CYP3A43 also produce the same two major metabolites [5,6].

Drug–drug interactions (DDIs) involving human CYP3A enzymes and alprazolam have repeatedly been reported in the past. For instance, excessive sedation can result from the concomitant administration of alprazolam with CYP3A4 and/or CYP3A5 inhibitors such as clarithromycin, erythromycin, itraconazole, ketoconazole, nefazodone, or ritonavir [7,8,9]. Other factors, such as inflammation or the intake of alcohol or grapefruit juice, may have similar consequences [7,10,11]. Moreover, CYP3A4 activities can vary significantly between individuals, CYP3A5 activity is absent in a significant part of the population, and CYP3A7 is only expressed in a small percentage of adults. Thus, at least in a subset of patients, the contribution of CYP3A43 to alprazolam metabolism may be more important than has been realized so far. In addition, CYP3A43 is by far the most understudied member of this P450 subfamily [12].

CYP3A43 was the fourth (and last) enzyme of the CYP3A family to be identified. It was only discovered in 2001 [13], at a time when other family members had already been known for decades. Therefore, there are comparatively few studies on this enzyme, and its crystal structure has not yet been resolved. There are reports on the differential effect of an intronic CYP3A43 polymorphism (rs472660) on olanzapine clearance in patients [14,15], and we were recently able to demonstrate by enzymatic assays that this antipsychotic is indeed a CYP3A43 substrate [16]. Another polymorphic gene variant (CYP3A43*3) contains the SNP rs680055 and codes for a mutated enzyme (CYP3A43:p.Pro340Ala), which is associated with an elevated risk for prostate cancer and also with increased mortality [17,18,19]. CYP3A43:p.Leu293Pro and CYP3A43:p.Thr409Arg are two artificial mutants whose creation was inspired by the CYP3A4.18 and CYP3A7.2 alleles, respectively [20], and which influence the enzyme’s activity towards olanzapine [16]. Apart from olanzapine, the only known drug metabolized by CYP3A43 is alprazolam (see above). However, this finding was only reported in a single study, and, moreover, nothing was known about the influence of CYP3A43 mutants on this activity. Therefore, it was the aim of this study to evaluate the consequences of such single-amino-acid changes on the enzyme’s properties. For this purpose, we first monitored alprazolam metabolism by all four CYP3A enzymes recombinantly expressed in our standard microbial model host for human CYP expression, the fission yeast *Schizosaccharomyces pombe* [21]. Then, we tested the three previously described CYP3A43 mutants mentioned above using the same method.

## 2. Materials and Methods

### 2.1. Fine Chemicals

Alprazolam and α-hydroxyalprazolam solutions were purchased from Sigma Aldrich Cerilliant^®^ (St. Louis, MO, USA); the NADPH generating system (solution A and B) from Corning GmbH (Amsterdam, Netherlands); acetonitrile and methanol from Merck (Darmstadt, Germany). Formic acid was obtained from Honeywell International Inc. (Bucharest, Romania); ethanol was purchased from Merck KGaA (Darmstadt, Germany); fresh ultrapure water was obtained from the water purification system LaboStarTM 2-DI/-UV, equipped with a Millipak 0.22 μm membrane point-of-use cartridge from Millipore (Merck, Darmstadt, Germany). All other chemicals and solvents used were of the highest grade available.

### 2.2. Fission Yeast Strains and Media

All the fission yeast strains used in this study have been described previously (all strains are listed in Table 1). The preparation of Edinburgh Minimal Medium (EMM) and basic manipulation methods of *S. pombe* were carried out as described [22]. Briefly, strains are normally fed in Edinburgh Minimal Medium (EMM) at 30 °C, with supplements at a final concentration of 0.1 g·L^−1^ when required. Liquid cultures were kept shaking for 36 h at 230 rpm and 30 °C. Thiamine was supplemented at a concentration of 5 μM.

### 2.3. Biotransformation with Permeabilized Fission Yeast Cells (Enzyme Bags)

Fission yeast strains were grown in a 10 mL liquid culture of EMM with supplements as needed at 30 °C and 230 rpm for 24 h, as described previously [25]. Briefly, for each assay, 5 × 10^7^ yeast cells were transferred to 1.5 mL Eppendorf tubes, pelleted and incubated in 1 mL of 0.3% Triton-X100 in Tris-KCl buffer (200 mM KCl and 100 mM Tris-HCl pH 7.8), and incubated at room temperature for 60 min at 210 rpm for permeabilization. Cells were washed three times with 1 mL of NH_4_HCO_3_ buffer (50 mM, pH 7.8), and then resuspended in 200 μL of reaction buffer containing alprazolam (500 µM), NADPH regeneration system (100 µM), and NH_4_HCO_3_ buffer (45 mM, pH 7.8). After that, the samples were incubated at 37 °C under 230 rpm for 24 h. For HPLC-MS analysis, samples were centrifuged at 5000× *g* for 10 min, the supernatant was collected, and the final liquid was stored at −20 °C until it was analyzed.

### 2.4. HPLC-MS Analysis

HPLC-MS conditions for the analysis and quantitation of alprazolam and its metabolites were performed on Agilent 6550 iFunnel QTOF attached to an Agilent 1290 II UHPLC (Santa Clara, CA, USA), equipped with a Zorbax Eclipse plus C18-column (10 cm × 2.1 mm, 1.8 μm). Elution was performed using formic acid in ultrapure water (0.1%) as mobile phase A and formic acid in acetonitrile (0.1%) as mobile phase B, at a flow rate of 0.25 mL/min, running a gradient (10% B to 45% B at 4.5 min, 1.5 min hold, 45% B to 90% B at 12 min, 2 min hold, back to 10% B at 14.1 min). The MS was operated in positive electrospray ionization mode at a capillary voltage of 4000 V and a nozzle voltage of 1000 V. A drying gas flow of 12 L/min at 200 °C, a sheath gas flow of 11 L/min at 350 °C, and a nebulizer pressure of 45 psi were used. Mass calibration was performed prior to analysis, using a calibration solution provided by the manufacturer. In MS/MS experiments, *m*/*z* 325 was used as the target precursor, and a collision energy of 19 eV was used to generate the product ions. The experiments were conducted in three repetitions, and three parallel samples were analyzed from each biotransformation reaction.

### 2.5. Statistical Analysis

All data are presented as mean ± SD. Statistical significance was determined using a two-tailed t-test. Differences were considered significant if *p* < 0.05. Statistical analysis was conducted using GraphPad Prism 5.01 (GraphPad Software, Inc., La Jolla, CA, USA).

### 2.6. Prediction of Oxidation Sites

Prediction of the site of oxidation (SoO) was conducted on the SMARTCyp server v. 3.0. SMARTCyp takes the 2D structure of a compound or builds it itself from a 1D representation (SMILES string) and calculates the energy required for the oxidation of each atom. It returns the most probable SoO under CYP catalysis.

### 2.7. Homology Modeling

Since CYP3A5 shows the highest preference for the oxidation of the methyl group (producing α-OH alprazolam), while CYP3A43 shows a strong preference for producing 4-OH alprazolam, we investigated different metabolization patterns of CYP3A5 and CYP3A43. Due to the unavailability of a public crystal structure for CYP3A43, homology modeling based on the structurally most similar conformation of CYP3A4, which bears the highest amino acid sequence homology to CYP3A43 (identity: 75.7%, similarity: 86.7%), was performed. We conducted our previously [16] established workflow of rationalized template selection. The co-crystallized CYP3A4 ligand most similar to alprazolam was detected, and the corresponding crystal structure was, consequently, together with the UniProt [26] identifier (Q9HB55-1) for CYP3A43, chosen as input for the SWISSMODEL [27] server to generate the homology model. The coordinates of the heme moiety were transferred from the CYP3A4 template to the CYP3A43 model. The CYP3A5 structure (PDB code: 7LAD [28]) was selected for modeling, as it is the only structure available in the Protein Data Bank (PDB) [29] with one co-crystallized ligand, thus indicating high comparability to the CYP3A43 model. The resulting model for CYP3A43 was curated and carefully checked in the MOE software package (Molecular Operating Environment 2020.0901, Chemical Computing Group ULC, Montreal, QC, Canada), including the setting of protonation states at pH 7.4 [30] in the OPLS-AA force field [31]. Dihedral angle outliers and steric clashes present in the CYP3A43 model were manually modeled using the MOE loop modeler tool and local energy minimizations.

### 2.8. Molecular Docking

Docking experiments for alprazolam binding to the CYP3A5 crystal structure (PDB code: 7LAD) and the CYP3A43 homology model mentioned above were conducted to study their possible interaction mode. Docking experiments were carried out using GOLD v. 5.8.1 (Genetic Optimization for Ligand Docking, CCDC Software, Cambridge, UK [32]). The genetic algorithm (GA) was run 60 times for each CYP enzyme, with searching efficiency of 200%. The docking site was set within an 18 Å radial sphere around the heme iron. The docking output conformations were energy-minimized in the MMFF94 force field [33] with LigandScout v. 4.4.3 (Inte:ligand, Vienna, Austria) [34]. Plausible docking poses were selected according to the distance between the SoO and Heme-Fe, supervised by the 6 Å rule of thumb [35], and the potential binding conformation of the ligands with the highest number of chemical interactions (H-bonds, lipophilic contacts, and charges) was calculated by LigandScout v. 4.4.3.

## 3. Results

The recombinant expression of human CYPs, together with their redox partner cytochrome P450 reductase (CPR), was carried out as described previously [23]. All of the fission yeast strains used in this study are listed in Table 1. The biotransformations were performed using enzyme bags (permeabilized fission yeast cells) as described before [25]. We first monitored the alprazolam metabolites produced by the four wild-type forms of the human CYP3A subfamily members (CYP3A4.1 to CYP3A43.1) and confirmed the production of both 4-hydroxyalprazolam and α-hydroxyalprazolam by each of them, as expected. Exemplary EIC and MS2 spectra are shown in the supplemental information (SI Appendix A). These results are in good agreement with the predictions of the CYP3A4 sites of alprazolam metabolism made by SMARTCyp [36], a web-based software that predicts the CYP-mediated metabolic liability of heavy substrate atoms (Appendix A). The metabolites M1 and M2 (4α/β-hydroxy alprazolam) showed the highest predicted probability, followed by M3 (α-hydroxy alprazolam) and M4 (5-N-O alprazolam). CYP3A5 (332 nM/d) and CYP3A4 (267 nM/d) showed the highest production rates of 4-hydroxyalprazolam, while CYP3A7 (88 nM/d) and CYP3A43 (16 nM/d) displayed lower activities (Figure 1). In the case of α-hydroxyalprazolam, CYP3A5 (710 nM/d) was by far the most effective producer, with CYP3A4 (34 nM/d), CYP3A7 (9.9 nM/d), and CYP3A43 (2.0 nM/d) all having much weaker activities. The comparison of these activity data with previous works is complicated by the fact that the four CYP3A enzymes have never been recombinantly expressed using the same approach. One study used COS-1 cells for the expression of CYP3A4 and CYP3A43 and reported similar production rates for both enzymes with respect to 4-hydroxyalprazolam; however, they did not see any production of α-hydroxyalprazolam by CYP3A4 at all [6].

Next, we monitored the alprazolam metabolites that are formed by the three previously described CYP3A43 mutants. Unexpectedly, all three mutants produced 4-hydroxy-but not α-hydroxyalprazolam. Instead, they all produced 5-N-O alprazolam. With respect to 4-hydroxyalprazolam, CYP3A43:p.Leu293Pro (107 nM/d) and CYP3A43:p.Pro340Ala (99 nM/d) showed sixfold higher production rates compared to the wild-type (CYP3A43.1), while the activity of CYP3A43:p.Thr409Arg (67 nM/d) was fourfold higher. By contrast, the production rates of 5-N-O alprazolam were all in the same range for the mutants (18 to 29 nM/d) and were not statistically significantly different from each other. In contrast, no detection of this metabolite was observed in the wild-type incubations. An overview of the different metabolic pathways is shown in Figure 2.

The CYP3A4 crystal structure with the co-crystallized ligand midazolam, a drug from the benzodiazepine class which is closely related to the analyzed drug alprazolam, turned out to be the most similar ligand measured by the TanimotoCombo (shape and color) score implemented in ROCS [37]. Consequently, the midazolam-bound crystal structure (PDB code: 5TE8) [38] served as a template for our CYP3A43 homology model. The resulting homology model showed one atom clash and two dihedral angle outliers located in residues not constituting the binding site, which were fixed manually by local minimization.

In order to investigate the isoforms’ metabolism capability for alprazolam, we chose the docking conformations of alprazolam in CYP3A5 and CYP3A43 as representative binding modes, as they have different preferences regarding the produced metabolites. Since midazolam is highly similar to alprazolam, we consider the key hydrogen bond between midazolam imidazole nitrogen and the Ser119 gamma hydroxy group in the CYP3A4-midazolam crystal structure (PDB code: 5TE8), discussed by Sevrioukova et al. [38] as an essential contributor to alprazolam binding by other CYP3As, too. Molecular dynamics simulations analyses conducted by Redhair et al. also monitored and assessed the frequent occurrence of Ser119-midazolam interaction in CYP3A4 [39]. The importance of Ser119 in CYP3A4 for the metabolism of alprazolam has been also stressed by Huang et al. [11], as they suggested ethanol-blocked Ser119 would cause an activity decrease in alprazolam metabolism.

CYP3A5 remarkably produces α-OH and 4-OH alprazolam, with a preference for the oxidation of the methyl group. Our computational experiments show a favorable alprazolam conformation for the production of α-OH alprazolam with the methyl group oriented towards the heme iron and within the required distance of 6 Å. This alprazolam binding mode is stabilized by the presumably important hydrogen bond between the triazole nitrogen and Ser119, as shown in Figure 3A, and features additional lipophilic contacts with the hydrophobic residues Ala370, Thr309, Val369, Leu481, and Leu305.

For CYP3A43, the conformation with C4 as SoO is shown in Figure 3B. Compared with the docking result for alprazolam in CYP3A5, and the crystal structure of midazolam in CYP3A4 (PDB code: 5TE8), here, in addition to the hydrogen bond between Ser119 and the ligand’s heterocycle nitrogen, another hydrogen bond between the Gln105 epsilon nitrogen of CYP3A43 and the neighboring triazole nitrogen of alprazolam can be seen in the binding hypothesis. It is worth noticing that this interaction with Gln105 cannot be easily reproduced with CYP3A5 due to its homologous Arg105, whose hydrogen bond donor positions are essentially oriented differently. This combination of two hydrogen bonds potentially stabilizes the triazole in the presented conformation. A factor that might influence the conformational difference between midazolam and alprazolam is the chlorine, as it may form a halogen bond to Pro481, whose corresponding residue in the other CYP3A enzymes is a glycine. Additional hydrophobic contacts are formed between alprazolam and Ala305, Thr309, Ile482, Val370, and Leu216.

## 4. Discussion

Previously, research by Sevrioukova et al. [38] shed light on the relevance of the residue Ser119 for the binding of the benzodiazepine drug midazolam to CYP3A4. The importance of Ser119 is further highlighted by the observation that mutating this residue precipitates drastic effects on the enzyme’s metabolic capabilities, regarding both overall turnover and selectivity between methyl and C4 oxidation [40]. For closely related CYP enzymes CYP3A5 and CYP3A43, there is evidence that the homologous Ser119 likewise contributes to the binding of alprazolam in a manner facilitating the metabolism to form α-OH and 4-OH alprazolam, respectively. In the case of CYP3A5, in addition to the predominant methyl oxidation, significant C4-hydroxylated product is also being generated. Therefore, Ser119 might act as an axis, holding the substrate in place while rotating, and thereby switching between conformations preferring C4 metabolism, thus enabling methyl metabolism conformation. This hypothesis will be further explored by prospective research endeavors.

## Figures and Tables

**Figure 1 biomedicines-10-03022-f001:**
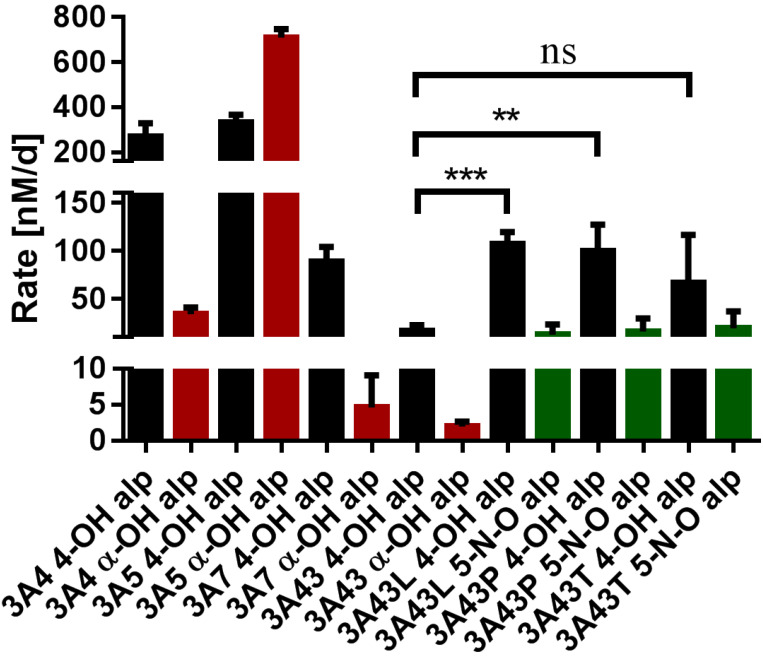
Production rates of the alprazolam metabolites 4-hydroxyalprazolam (4-OH alp; black columns), α-hydroxyalprazolam (α-OH alp; red columns), and 5-N-O alprazolam (5-N-O alp; green columns) after biotransformation with recombinant human CYP3A enzymes as indicated; ** *p* < 0.01; *** *p* < 0.005 versus control (3A43 4-OH alp). The CYP3A43 mutants are CYP3A43:p.Leu293Pro (3A43L), CYP3A43:p.Pro340Ala (3A43P), and CYP3A43:p.Thr409Arg (3A43T), respectively.

**Figure 2 biomedicines-10-03022-f002:**
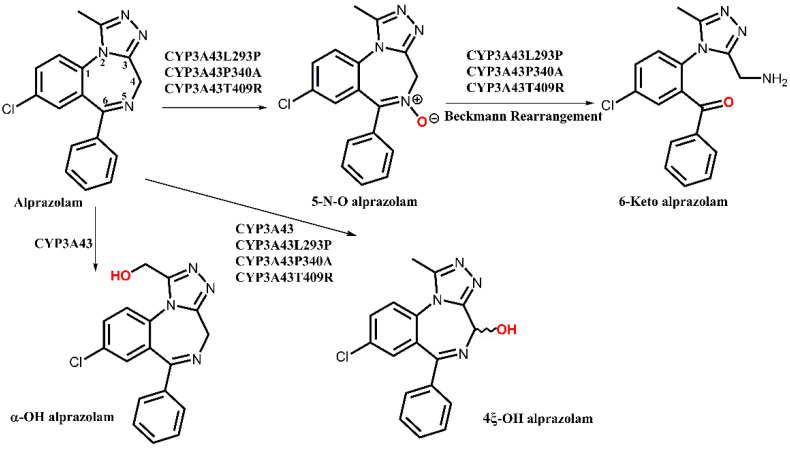
Postulated metabolic pathways of alprazolam by human CYP3A43.1 and its mutants as indicated.

**Figure 3 biomedicines-10-03022-f003:**
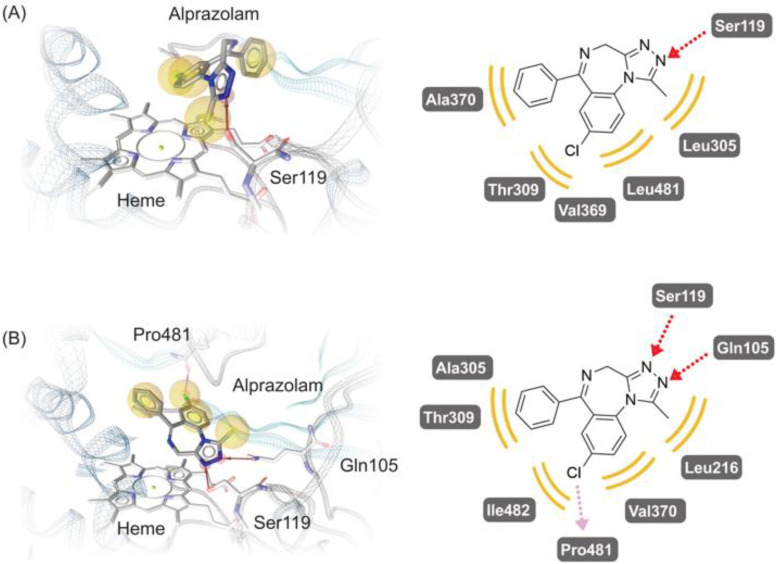
Suggested docking conformations of alprazolam as substrate of CYP3A5 (**A**) and CYP3A43 (**B**). Interactions between alprazolam and the studied CYPs are presented in 3D pharmacophore (**left**) and their 2D representation (**right**). In both 3D and 2D pharmacophore schemes, red arrows stand for hydrogen bond donors, the violet arrows stand for halogen bonds, and the yellow spheres represent hydrophobic contacts.

**Table 1 biomedicines-10-03022-t001:** Fission yeast strains used in this study.

Strain	Parental Strain	Expressed Protein(s)	Genotype	Reference
**CAD62**	NCYC2036	hCPR	h-ura4-D.18 leu1::pCAD1-CPR	[23]
**CAD67**	CAD62	hCPR, CYP3A4.1	h-ura4-D.18 leu1::pCAD1-CPR/pREP1-CYP3A4*1	[23]
**INA20**	CAD62	hCPR, CYP3A5.1	h-ura4-D.18 leu1:: pCAD1-CPR/pREP1-CYP3A5*1	[20]
**INA2**	CAD62	hCPR, CYP3A7.1	h-ura4-D.18 leu1:: pCAD1-CPR/pREP1-CYP3A7*1	[24]
**INA43**	CAD62	hCPR, CYP3A43.1	h-ura4-D.18 leu1::pCAD1-CPR/pREP1-CYP3A43*1	[21]
**ZJ1**	CAD62	hCPR, CYP3A43:p.Leu293Pro	h-ura4-D.18 leu1::pCAD1-CPR/pREP1-CYP3A43-L293P	[16]
**ZJ2**	CAD62	hCPR, CYP3A43:p.Pro340Ala	h-ura4-D.18 leu1::pCAD1-CPR/pREP1-CYP3A43*3	[16]
**ZJ3**	CAD62	hCPR, CYP3A43:p.Thr409Arg	h-ura4-D.18 leu1::pCAD1-CPR/pREP1-CYP3A43-T409R	[16]

## Data Availability

Not applicable.

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
