# Peer review of "Changes in Alprazolam Metabolism by CYP3A43 Mutants"

_biomedicines, 2022, doi:10.3390/biomedicines10123022_

Round 1
Reviewer 1 Report
In the article entitled „Changes in alprazolam metabolism by CYP3A43 mutants”, the authors have done an excellent studying and explaining the difference in alprazolam metabolism by different CYP3A enzymes. I have just one comment:
1. The authors did not really explain the influence of the three CYP3A43 mutations (L293P, T409R, and P340A). The role of Ser119 and Gln105 is discussed, however the three mutations are not. I would really want to know the authors thoughts on this.
Reviewer 2 Report
This study titled 'Changes in alprazolam metabolism by CYP3A43 mutants" present important findings relating to the effects of polymorphism on the metabolic activity of CYP3A43. The results add to the current body of knowledge in this field and can help unearth several other unknown issues around the CYP3A family.
The design of the study is generally sound and follows the template of similar studies. The use of alprazolam as probe substrate of CYP3A, along with the monitoring for the alpha and 4-hydroxy metabolites are well established.
Some comments
1. The third paragraph of the introduction looks like a misfit. It does not appear relevant to the study. CYP3A43 plays roles in the metabolism of some antipsychotic agents. That is sufficient to say. What is missing is an in-depth presentation of CYP3A43, its polymorphism and potential implication for drug therapy. Also missing in the introduction is the scientific basis/justification of study, choice of methodology (e.g., alprazolam vs other substrates, why not olanzepine? mutant expression methodology, etc)
2. Are there any limitations to this findings, especially because of the use of, and reliance on docking to predict the metabolic activity of the isoforms? Are the mutants commercially available?
3. Homology modeling of CYP3A43 was performed using CYP3A4 information. What does the 75.5% identity and 86.7% similarity between CYP3A4 and CYP3A43 mean? Is this activity, amino-acid sequence, or what? since the crystal structure of CYP3A43 is not publicly available.
